# Perceptual Structure in the Absence of Grounding for LLMs: The Impact of Abstractedness and Subjectivity in Color Language

**Pablo Loyola[1], Edison Marrese-Taylor[2,3], Andres Hoyos-Idobro[4]**

[1] Rakuten Institute of Technology; Rakuten Group, Inc; Tokyo, Japan
[2] National Institute of Advanced Industrial Science and Technology, Japan
[3] Graduate School of Engineering, The University of Tokyo
[4] Rakuten Institute of Technology; Rakuten Group, Inc; Paris, France
{pablo.a.loyola,andres.hoyosidrobo}@rakuten.com, emarrese@weblab.t.u-tokyo.ac.jp

## Abstract

The need for grounding in language understanding is an active research topic. Previous work has suggested that color perception and color language appear as a suitable test bed to empirically study the problem, given its cognitive significance and showing that there is considerable alignment between a defined color space and the feature space defined by a language model. To further study this issue, we collect a large scale source of colors and their descriptions, containing almost a 1 million examples , and perform an empirical analysis to compare two kinds of alignments: (i) inter-space, by learning a mapping between embedding space and color space, and (ii) intra-space, by means of prompting comparatives between color descriptions. Our results show that while color space alignment holds for monolexemic, highly pragmatic color descriptions, this alignment drops considerably in the presence of examples that exhibit elements of real linguistic usage such as subjectivity and abstractedness, suggesting that grounding may be required in such cases.

## 1 Introduction

One of the most interesting aspects of Large Language Models (LLMs) is that while they are usually trained without explicit linguistic supervision, or knowledge injection, there is a relevant body of work that shows that both linguistic structures and relational knowledge emerge even without the need for fine-tuning (Petroni et al., 2019; Goldberg, 2019; Marvin and Linzen, 2018). This has generated an ongoing discussion on how these models are learning and if the current training objectives and text-only data are enough to cover a wide range of downstream tasks.

One of the perspectives that have been used recently to study this phenomenon is color language (Abdou et al., 2021). Color naming is a relevant task (Steels et al., 2005), well understood in physiological (Loreto et al., 2012) and sociocultural

terms (Gibson et al., 2017) and has an inherent intent from the communicative point of view (Zaslavsky et al., 2019; Twomey et al., 2021). In a recent work, Abdou et al. (2021) proposed to quantify the *alignment*, understood as the structural correspondence between a point in the color space (e.g. RGB or CIELAB) and its associated name in natural language represented as the feature vector obtained from a LLM. For such empirical study, they make use of the Color Lexicon of American English (Lindsey and Brown, 2014), based on the Munsell Chart of color chips (Munsell et al., 1915), finding that in general alignment is present across the color spectrum. While this work provides valuable insights on the actual grounding requirements, most of the color descriptions are monolexemic.

The above findings sparked our interest in the issue and led us to run preliminary experiments to test to what degree this alignment exists for less pragmatic color descriptions. We observed that such alignment drops significantly in the presence of more complex and subjective ways to describe specific colors, for example, when the color names contain multiple nouns or NPs, as well as terms with other parts-of-speech.

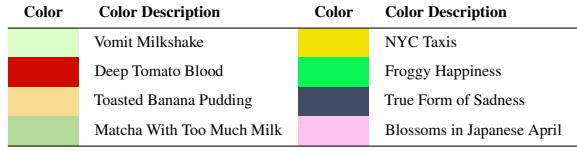

| Color | Color Description | Color | Color Description |
|---|---|---|---|
| | Vomit Milkshake | | NYC Taxis |
| | Deep Tomato Blood | | Froggy Happiness |
| | Toasted Banana Pudding | | True Form of Sadness |
| | Matcha With Too Much Milk | | Blossoms in Japanese April |

Table 1: Selection of samples from the COLORNAMES dataset, showing the richness of color descriptions.

To further study these issues, we construct a more challenging test scenario by using and processing data from ColorNames[1], an online service where users can collaboratively generate (color, color description) pairs. Given its free-form structure, the COLORNAMES dataset represents a rich

---

[1] https://colornames.org

and heterogeneous corpus of color descriptions. This can be appreciated clearly in Table 1.

Using this new, more challenging dataset, we conducted two experiments. The first one complements the work of Abdou et al. (2021) in assessing the inter-space alignment between color and LLM spaces, with two key new additions: (i) we propose a more fine-grained color name segmentation, considering metrics associated to subjectivity and concreteness, and (ii) we adopt an Optimal Transport-based metric to complement the existing alignment methods. For the second experiment, we focus on the representations obtained from the LLMs and their ability to ground color comparatives as a way to structure color descriptions. Critically, we do this without the need of accessing underlying points in color space. Concretely, we assess to what extent the LLM is able to discover a comparative-based relationship between a pair of color names without the need for explicit underlying color information, following a few shot learning configuration. For example, what would be the correct comparative associated to the pair of color names (e.g. between *blood red wine* and *funeral roses*)? If the model succeeds on that task, it could mean that such relationships are somehow encoded during pretraining, without the need of an explicit color signal.

The results of the proposed experiments show in general the alignment scores between spaces on the proposed dataset are low, contrasting with the results provided on (Abdou et al., 2021) on the Munsell dataset. This means that the complexity of the color descriptions, exemplified by the subjectivity and concreteness metrics, really impact on the perceptual structure the language models achieve. On the other hand, the results of the second experiment on comparative prediction, show that all language models are able to perform surprisingly well, even in scenarios of high subjectivity. This discrepancy leads to think that somehow the models retain certain structure learned thought language, but are not able to translate into color modality.

## 2 Related Work

Color language, as a medium to study grounding, has been used in several works, mainly trying to learn a mapping between the descriptions and their associated points in color space, such as Kawakami et al. (2016), based on character level model, Monroe et al. (2016); McMahan and Stone (2015) which

take as input a color representation and generates a natural language description, Monroe et al. (2017), who incorporates the idea of contextual information to guide the generation, and in Monroe et al. (2018) tested it in a bilingual setting. Winn and Muresan (2018); Han et al. (2019) proposed to model comparatives between colors. In most of these works, the source of color names come from Munroe (2010), which compresses the results of an online survey where participants were asked to provide free-form labels in natural language to various RGB samples. This data was subsequently filtered by McMahan and Stone (2015), leading to a total number of samples of over 2 million instances, but with a number of unique color names constrained to only 829. This suggests a reduction in the complexity of modeling tasks as proposed by previous work, as the vocabulary is fairly small, and with a homogeneous frequency. In contrast, the empirical study we propose does not have such constraint, allowing us to work with unique, subjective descriptions which are richer in vocabulary. In terms of using color to understand perceptual structure in LLMs, our direct inspiration was the work by Abdou et al. (2021), where authors perform experiments to quantify the alignment between points in color space and embeddings obtained by encoding the associated color names with LLMs.

## 3 Experimental Setting

**Data:** We use data from ColorNames[2], which crowdsources (color, color name) pairs. Table 1 presents some examples. The extracted data was filtered to standardize the comparison. Only English sentences were kept, spam entries were removed using predefined rules and only color names with a maximum of five words were kept. The resulting dataset consists of 953,522 pair instances, with a total vocabulary size of 111,531 tokens. As seen in Table 2, words with the highest frequencies correspond to *color words*. In terms of POS patterns, the data presents a total of 3,809 combinations, extracted using Spacy[3] but the most frequent patterns represent ways to modify nouns, by using an adjective (e.g. *dark apple*) or a sequence of nouns. We computed *concreteness* scores for the descriptions based on Brysbaert et al. (2014), which provides a defined set of lemmas with a ranking varying from 1 to 5. For example, *red pine brown* gets a score

---

[2]colornames.org
[3]https://spacy.io/usage/linguistic-features

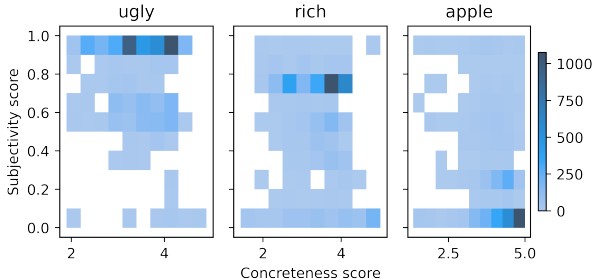

Figure 1: Joint histograms showing subjectivity and concreteness scores for color descriptions containing the terms *ugly*, *rich* and *apple*, respectively.

| Word | Frequency | POS Pattern | Frequency |
|---|---|---|---|
| green | 56,542 | ('ADJ', 'NOUN') | 119,752 |
| purple | 40,272 | ('NOUN', 'NOUN') | 78,678 |
| blue | 29,880 | ('VERB', 'NOUN') | 64,720 |
| pink | 20,878 | ('PROPN', 'NOUN') | 55,142 |
| red | 17,369 | ('ADJ', 'NOUN', 'NOUN') | 53,349 |
| yellow | 15,135 | ('PROPN', 'PROPN') | 42,701 |

Table 2: Most common words and most common POS patterns across color descriptions.

of 4.3, while *mysterious skyscape* gets a score of 1.9. In this sense, we make the assumption that lower concreteness means higher abstractedness. Additionally, *subjectivity* scores were computed based on TextBlob (Loria et al., 2018), a rule-based approach that provides a score between 0 and 1, where a description like *thick and creamy* gets a score of 0.47, and *mathematically perfect purple* gets a 1.0. Figure 3 shows the correspondence between the scores and the expected usage of three sample words, ranging from *ugly*, a term associated with subjectivity to *apple*, which is commonly used to represent the *reds* spectrum. In the case of *rich*, it could have mixed connotations, which is reflected in its histogram having most frequent values at the center.

**Language Models:** For all experiments, we used three language models, namely, BERT (Devlin et al., 2019), Roberta (Liu et al., 2019), T5 (Raffel et al., 2020), all of them in their *large* type, based on their availability on HuggingFace[4]. As a control baseline, we use FastText word embeddings (Bojanowski et al., 2017). The selection of such models is based on the need to explicitly extract embeddings from the LLM, which means that in principle only white-box models can be used. This ruled out API-based systems like GPT-3 and other similar. Moreover, even for competitive white-box models such as LLaMa, given the size of our introduced dataset (around million examples), its usage is left for future work. Finally, we note that our selection of LLMs lies within the masked-language modelling domain (MLM). This is a deliberate and critical decision, as it allows for our experiments to be performed in a controlled in-filling setting, limiting what the LLM can output and allowing us to parse their generations automatically. More

---

[4] https://huggingface.co/models

up-to-date models are all causal LMs (with a few exceptions), which means that our capacity to control is more limited, as these models cannot in-fill text. Moreover, it has been shown that the output of these models is highly dependent on how they are prompted, usually requiring a huge amount of work into prompt construction in order to control the output, which adds further complications.

## 4 Experiments

**Experiment I: Inter-space Alignment** This first experiment is directly inspired by Abdou et al. (2021). In this case, we want to assess the alignment between color and LM feature spaces. For measuring such alignment, we replicated the settings proposed by (Abdou et al., 2021) for the case of (i) **Linear Mapping (LMap)**, where given a set of $n$ (color, color name) pairs, the alignment is measured as the fitness of the regressor $\mathbf{W} \in \mathbb{R}^{d_{\text{LM}} \times 3}$ that minimizes $||\mathbf{XW} - \mathbf{Y}||_2^2 + \alpha||\mathbf{W}||_1$, with $\alpha$ regularization parameter, $\mathbf{X} \in \mathbb{R}^{n \times d_{\text{LM}}}$ the color names embeddings and $\mathbf{Y} \in \mathbb{R}^{n \times 3}$ the vectors coming from the color space, and (ii) **Representational Similarity Analysis (RSA)** (Kriegeskorte et al., 2008), the non-parametric method, whose score is operationalized via the mean Kendall's $\tau$ between both modalities. In addition, we propose to model alignment as an **Optimal Transport (OT)** (Peyré et al., 2019) problem, where the goal is to find a transport matrix that minimizes the cost of moving all text samples onto their corresponding color samples. We rely on Gromov-Wasserstein distance (GW) (Peyré et al., 2016), which extends the OT setting from samples to metric spaces. GW finds a mapping $\mathbf{T} \in \mathbb{R}_+^{n \times n}$ that minimizes the GW cost, $\sum_{i,j,k,l} ||\mathbf{C}_{ik}^{\text{TEXT}} - \mathbf{C}_{jl}^{\text{COLOR}}||_2^2 \mathbf{T}_{ij} \mathbf{T}_{kl}$ subject to $0 \leq \mathbf{T}_{ij} \leq 1$, $\sum_i \mathbf{T}_{ij} = \frac{1}{n}$, and $\sum_j \mathbf{T}_{ij} = \frac{1}{n}$, where $\mathbf{C}^{\text{TEXT}} = \cos(\mathbf{X}, \mathbf{X}) \in \mathbb{R}^{n \times n}$ and $\mathbf{C}^{\text{COLOR}} = \cos(\mathbf{Y}, \mathbf{Y}) \in \mathbb{R}^{n \times n}$ are the within-domain similarity matrices. Therefore, $\mathbf{T}_{ij}$ denotes the probability of assigning a sample $i$ in the text space to a sample $j$ in the color space.

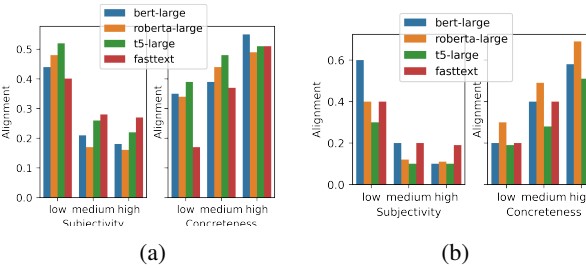

(a)             (b)

Figure 2: Alignment using (a) LMap (b) OT.

| | Reference | Comparative | Target | |
|---|---|---|---|---|
| | Rotten meat red | DARKER than | Ripe red pepper | |
| | Light forest greenish | LIGHTER than | Dead field green | |
| | Stretchy sweater pink | DEEPER than | Chewed bubblegum pink | |

Table 3: Examples of color descriptions from COLOR-NAMES and their most suitable comparative (Winn and Muresan, 2018) as obtained by our matching procedure.

We grouped the examples into uniform segments associated to the subjectivity and concreteness scores, computing alignment scores per segment, per LM. In general, results show the alignment is low, and it decreases as subjectivity increases. Similarly to (Abdou et al., 2021), FastText behaves as a strong baseline, we hypothesize that given the uniqueness of the descriptions, its n-gram based tokenizer could be more stable than the tokenizers implemented within the LMs. Figure 2 show the results for all models on all segments using LMap and OT. (We omitted RSA as its behavior is very similar to LMap).

**Experiment II: Perceptual Structure via Comparative Identification** The objective of this experiment is to determine if a LM can structure relationships between color descriptions without accessing the associated points in color space. To that end we design a task where, given two color descriptions, the LM has to determine the correct *comparative* that relates both descriptions (e.g. *darker* or *lighter*). To this end, we firstly match the dataset provided by Winn and Muresan (2018), which consists of tuples ( [ *reference color points* ], *comparative*, [*target color points*]) against COL-ORNAMES , by sampling (color, color description) pairs $(c_i, cd_i)$, $(c_j, cd_j)$ and retrieving the comparative that minimizes simultaneously the distance between [*reference color points*] and $c_i$, and [*target color points*] and $c_j$. After this step, we have, for any pair of descriptions in the COLORNAMES dataset, a ranking with the most suitable comparatives, based on explicit grounding. Table 3 provides matched examples.

We operationalize the task as a few shot inference. Firstly, we select randomly from the matched dataset, $K$ tuples of the form (*description i*, *comparative*, *description j* ), from which $K - 1$ are used to construct the labeled part of a prompt, following the template "$description_i$ is [compar-

ative] than $description_j$". The remaining $k$-th tuple is appended as "$description_i$ is [MASK] than $description_j$", i.e., the comparative has been masked. The resulting prompt is passed through the LM and the ranking of the most likely tokens for [MASK] is retrieved. As evaluation metric, we chose the Mean Reciprocal Rank (MRR), as it encodes the position of the correct answer among the raking provided by the LM. We experimented using a total set of 81 comparatives and varying parameter $K$ from 5 to 20. We performed the same uniform segmentation in terms of subjectivity and concreteness. Results in general showed surprisingly good results in terms of MRR, in several cases, LM outputs the correct comparative at position 1. There was a natural decay in performance when $K$ was smaller, but for $K > 10$ results were consistent across models and segments. Figure 3 presents the results for $K = 10$, showing uniformity that led us to speculate that for this task, subjectivity may not be relevant as a control factor. From a qualitative point of view, Figure 4 shows the result of constructing a graph based on the comparative relationships correctly inferred by the LM. As it can be seen, (a) there is color coherence in terms of the neighboring relationships, and (b) when sampling paths form the graphs, transitions are consistent. Further investigation of these structures is left for future work. Additionally, trying to assess the impact on the language model selection, we experimented considering ChatGPT (web-based query) and Llama-2 (llama-2-13b-chat) as language models tasked to predict the comparatives. We found that, in several cases (with different prompts) even when using a $k$-shot setting, these models often created new types of comparatives (e.g. "SIMILAR IN LIGHTNESS"). As such new comparatives are not present in the ground truth of our dataset, our evaluation framework becomes increasingly complicated since we would need to further post-process the model outputs. Such a more complex experiment is left for future work.

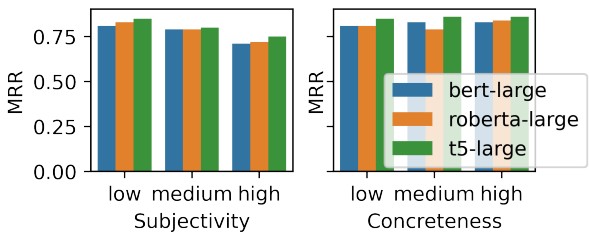

Figure 3: Mean Reciprocal Rank (MRR) for comparative inference across language models.

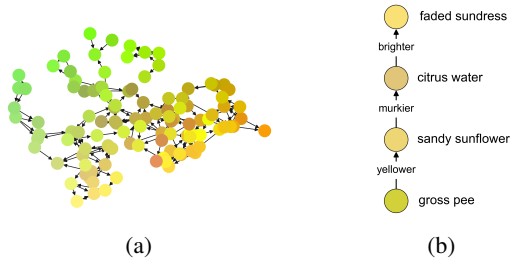

Figure 4: Examples of perceptual structure obtained by leveraging the correct comparative predictions.

| | taxi | sun | banana | taxi sun | taxi banana | sun banana | all |
|---|---|---|---|---|---|---|---|
| **Alignment** | 0.529 | 0.811 | 0.737 | 0.503 | 0.413 | 0.447 | 0.453 |
| **MRR** | 0.770 | 0.797 | 0.881 | 0.735 | 0.801 | 0.693 | 0.681 |

Table 4: Results for the sample extracted from the *yellow* spectrum of the dataset.

**Impact of inner context** Finally, differently from (Abdou et al., 2021), as our descriptions are not appended to any additional context at encoding time, we want to assess if their complexity acts as a natural source of context. For example, for the description *mustard taxi yellow*, we can see how the generic *yellow* color word is being conditioned by two nouns, *mustard* and *taxi*. An analysis of our data showed that out of 900K instances, around 390K (43 %) contain a color word. Based on this, we split the data into two chunks and re-run the two experiments described above. The results show that for the case of the alignment task, the mean R-scores from the regressor associated with the set of descriptions that have and do not have a color word are 0.401 and 0.381 respectively (using BERT-large). We can see that there is in fact a difference, although the ranges are within the results reported.

Moreover, given the full set of (color, color description) pairs, we cluster them using the color representations using k-means. From the resulting set of clusters, we choose the one that groups the *yellows*, as an example. Within that cluster, we performed a new grouping, this time using the embeddings of the color descriptions. From this, we now obtained 22 subgroups (again, using standard k-means) that are semantically close (inner context) but that are globally constrained by the color spectrum chosen (*yellow*). We now study the alignment within semantically-close groups and in-between

groups. We selected three distinct groups: a group with semantics about *taxis*, one about *bananas* and one about *sun/sunset/sunrise*. We computed the alignment score and the MRR associated to comparative prediction of each group independently, for pairs of groups and for the combination of all of them, as seen in Table 4.

As we can see, in general the alignment scores are reasonable for single groups, but as we start combining them, the scores drop. This is expected as in most cases there is a token that becomes an anchor which is slightly modified by the rest of the description. On the other hand, in the prediction of the comparative among pairs of descriptions, we can see that the accuracies (measured with MRR) dropping as we combine the sets, but still remain mostly in the same ballpark. This, while not conclusive evidence, helps us approximate to the notion that alignment can indeed be influenced by the semantics of the descriptions, but it does not seem to play a big role on how the LM structure the information using comparatives.

## 5 Conclusion and Future Work

We studied how LLMs encode perceptual structure in terms of the alignment between color and text embedding spaces and the inference of comparative relationships in a new, challenging dataset that encapsulates elements of real language usage, such abstractedness and subjectivity. The results show that LMs perform in mixed way, which provides additional evidence on the need for actual grounding. In terms of future work, we are considering the need for additional contextual information, which could be attacked by considering color palettes instead of single colors, and also considering a multilingual approach, to cover the sociocultural aspects of color naming, specially in scenarios of low resource translation.

## Acknowledgements

We would like to thank the anonymous reviewers for their insightful and constructive feedback.

## Limitations

One of the key limitations of the current work is its focus solely on English language, which, in terms of color language, naturally compresses the cultural aspects associated to English-speaking societies. This is clear when we analyze in detail the vocabulary, we can find cultural archetypes that are probably not transferable. In that, there is an inherent challenge on how to learn color naming conventions in a more broad way. For example, for applications related to low resource languages, understanding the use of language for color description could be helpful for anchoring linguistic patterns.

## Ethics Statement

Our main objective is to understand how language is used to communicate color. This has several applications, for example in e-commerce, such as search, product reviews (where color is an important attribute). While directly our study tries to abstract from specific user information, it is certain that language usage identification could be used for prospecting or targeting individuals from a specific cultural background.

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
