# OpenReview forum: "Perceptual Structure in the absence of grounding: the impact of abstractedness and subjectivity in color language for LLMs"
_EMNLP/2023/Conference — EMNLP 2023 Findings_

### Official Review · Reviewer_T8is · 2023-07-31

**Soundness:** 2

**Excitement:**

3: Ambivalent: It has merits (e.g., it reports state-of-the-art results, the idea is nice), but there are key weaknesses (e.g., it describes incremental work), and it can significantly benefit from another round of revision. However, I won't object to accepting it if my co-reviewers champion it.

**Paper Topic And Main Contributions:**

This paper contributes to the literature on the alignment between language models' color space and the visual properties of the color (i.e. comparing embedding distance to distance in RGB space). Whereas earlier work examined the embeddings of monoleximic color terms such as "red" or "yellow", this paper looks at more complex ad-hoc color names from an online database; those include "NYC taxis" and "vomit milkshake". The experiments reported in the paper found that alignment for these more complex color names was poor, but the model was able to order color terms by darkness, e.g., "rotten meat red" is darker than "ripe red pepper".

**Questions For The Authors:**

I must have missed this - how did you compute embeddings for multiword phrases? Averaged the embeddings of all of the subwords the make up the phrase?

**Reasons To Accept:**

The scientific question of how much language models can learn about colors without being exposed to visual information is an interesting one.

The paper is generally well written.

**Reasons To Reject:**

I may be fundamentally misunderstanding something, but the premise of the first experiment seems strange to me. Do we really expect the embedding of "NYC taxis" to be as close to the embedding of "vomit milkshake" as these colors happen to be close to each other in RGB space? At the risk of stating the obvious - unlike "yellow" and "red", which are pure color terms, the phrase "NYC taxis" is expected to carry a lot of semantic information that has nothing to do with color. Perhaps the experiment would make more sense if there was a way to prompt the model to focus on the colors denoted by those phrases - this is probably how you would set up a human experiment here? Speaking of human experiments, it would be helpful to get a sense of how well humans do on these tasks - not just for the similarity alignment task (where you'd have the person separately give similariy judgments to colors and to phrases like "vomit milkshare"), but also for the darkness rating task. I'm not personally sure, for example, that I would rate "stretchy sweater pink" as "deeper" than "chewed bubblegum pink".

**Reproducibility:**

4: Could mostly reproduce the results, but there may be some variation because of sample variance or minor variations in their interpretation of the protocol or method.

**Reviewer Confidence:**

3: Pretty sure, but there's a chance I missed something. Although I have a good feel for this area in general, I did not carefully check the paper's details, e.g., the math, experimental design, or novelty.

---

> ### Author Rebuttal · Authors · 2023-08-29
>
> We thank reviewer `T8is` for their comprehensive review. We appreciate their comments regarding our lack of clarity in terms of the scope and underlying assumption. We agree that things can be improved in this sense, and commit to doing so for the camera-ready version.
>
> Regarding their comments about the validity of comparing sentences such as `NYC Taxi` and `vomit milkshake`, we agree that without providing any additional context, there is little reason to expect that a model or (probably)
> a human would consider them as semantically related. Now, we would like to emphasize that our work is constrained to the field of color descriptions (as an instance of studying language grounding). What we are trying to do is to assess if these models, trained with the standard regimes, are able to internally structure perceptual information (in this case, color) that is communicated in terms of text-only input i.e. without explicitly receiving color input. We do so by means of the two experiments described in the paper.
>
> We think their suggestion of explicitly informing the LLM that we are considering a color description scenario is extremely pertinent and well-aligned with the comment above. Now, if we take a look at how Abdou et al. (2021) dealt with this issue, we can see that since their color names are monolexemic, they were able to match against the feature production norms dataset by McReid et al. (2005), where concepts (usually nouns) are linked to a series of features, one of them being the main color. For example, they have tuples such as (red, apple), (blue, sky). Given the simplicity of this dataset, Abdou et al. (2021) are able to use those tuples to generate template-based contexts such as `[apple] is [red]`. From this, one can clearly see how passing such a sentence to the LM could reinforce the notion that `red` in this context is used as a description.
>
> Differently from the above scenario, our descriptions are more complex, i.e., there are no canonical objects that have such colors descriptions as attributes, therefore the technique by Abdou et al. (2021) cannot be used. However, as we can see from the nature of our dataset, the richness of the descriptions can itself act as context. For example, for the description `mustard taxi yellow`, we can see how the generic `yellow` color word is being conditioned by two nouns, `mustard` and `taxi`. Note that such description contains the color word `yellow`. We could say that this is approximately what the template-based approach of Abdou et al. (2021)
> An analysis of our data showed us that out of 900K instances, around 390K (43 %) contain a ``color word''. Based on this, we split the data into two chunks and re-run the experiments from our paper. The results show that for the case of the alignment task, the mean R scores from the regressor associated with the set of descriptions that have and do not have a color word are 0.401 and 0.381 respectively (using BERT-large). We can see that there is in fact a difference, although the ranges are within the results reported in our paper.
>
> Moreover, in order to more comprehensively approximate an answer to your question, we performed one additional experiment. Given the full set of (color, description) pairs $D$, we cluster them using the color representations using k-means. From the resulting set of clusters $C$, we choose one, for example, the one that groups `yellows` $c_{yellow} \in C$. Within that set of (color, description) pairs, we can now perform a new grouping, this time using the embeddings of the descriptions. With this, we now obtain 22 subgroups (again, using standard k-means) that are semantically close (inner context) but that are globally constrained by the color spectrum chosen (yellow). For example, we see that there is a subgroup that contains most descriptions about taxis :
>
> | **color (hex)** | **description**        |
> |-----------------|------------------------|
> | #e9c503          | mustard taxi yellow    |
> | #f0d654          | mustard taxi cab       |
> | #b77a0b          | australian taxi yellow |
> | #c59813          | old taxi cab           |
> | #cab65e          | faded taxi bus         |
> | #f2b312          | plain taxi yellow      |
>
> If the reviewer is kind enough to check the hex colors in the table (can be input directly at https://colornames.org/), they will see that most colors are yellow-ish. We can now study the alignment within semantically-close groups and in-between groups. We selected three distinct groups: a group with semantics about `taxis`, one about `bananas` and one about `sun/sunset/sunrise`. We computed the alignment score of each group independently, for pairs of groups and for the combination of all of them:
>
>
> |           | `taxi` | `sun` | `banana` | `taxi-sun` | `taxi-banana` | `sun-banana` | `all` |
> |-----------|--------|-------|----------|------------|---------------|--------------|-------|
> | alignment | 0.529  | 0.811 | 0.737    | 0.503      | 0.413         | 0.447        | 0.453 |
>
>
> As we can see, in general the alignment scores are reasonable for single groups, but as we start combining them, the scores drop. This is expected as in most cases there is a token that becomes an anchor which is slightly modified by the rest of the description. On the other hand, if we conduct the prediction of the comparative among pairs of description (second experiment in the original paper), we can see that the accuracies (measured with MRR) dropping as we combine the sets, but still remain mostly in the same ballpark:
>
>
> |     | `taxi` | `sun` | `banana` | `taxi-sun` | `taxi-banana` | `sun-banana` | `all` |
> |-----|--------|-------|----------|------------|---------------|--------------|-------|
> | mrr | 0.770  | 0.797 | 0.881    | 0.735      | 0.801         | 0.693        | 0.681 |
>
> This, while not conclusive evidence, helps us approximate to the notion that alignment can indeed be influenced by the semantics of the descriptions, but it does not seem to play a big role on how the LM structure the information using comparatives.
>
> Finally, we also consider the reviewer has an important point regarding the need for human signals to provide more consistency to the results. While we consider LMs may provide general agreement, you are also right that depending on human factors, the perception of color can vary. That is a study we consider is currently out of the scope of a short paper, but for sure we consider it to include in future work, acknowledging cultural and demographic differences in color perception and color communication.

---

### Official Review · Reviewer_H3n7 · 2023-08-04

**Soundness:** 4

**Excitement:**

4: Strong: This paper deepens the understanding of some phenomenon or lowers the barriers to an existing research direction.

**Paper Topic And Main Contributions:**

Color Language has recently be used to investigate how and what LLMs learn in the pre-training phase.
For the purpose of studying this, they collect data from ColorNames, compiling a dataset of really creative and exotic color descriptions.
The first task is assessing the inter-space alignment between color and LLM spaces with fine-grained color name segmentation. For the second experiment, they focus on the representation obtained from the LLMs and their ability to match color comparatives with the structure of color descriptions.
They build upon the work of Abdou et al. (2021), which means this work is incremental.

Results show that the alignment scores are low, but results on comparative estimations are comparably high. The authors’ conclude that LLMs learn structure through language, but cannot transfer it to knowledge source-related concepts (such as color in this case).

For experiments, the authors used BERT, Roberta and T5  (all large)

In detail, the first experiment tries to align the positions of colours in colour space with their prose description in LLMs representational space. FastText embeddings are used as a baseline space.
Results for this experiment are low, performance is anti-proportional to the subjectivity of the expression.

The second experiment explores whether LLMs can structure relationships without the information given by the Color space. The task of the LLM is do determine, given two color descriptions, the correct comparative (e.g. whether color y is lighter or darker than y). With consistent results across data points, which are “surprisingly good”, the authors’ conclude that subjectivity is less important in this task.

**Reasons To Accept:**

This seems to be a good incremental contribution to an important topic, namely exploring the knowledge-related capacities of large language models. The paper is written with great clarity, all metrics are explained in detail.

**Reasons To Reject:**

The only criticism I could list here is that no more recent LLM like GPT-3.5 / 4 was used, but this is not the authors' fault
but dependent on the lack of availability of the models. Though, I am sure, these models may have performed significantly differently.

**Reproducibility:**

4: Could mostly reproduce the results, but there may be some variation because of sample variance or minor variations in their interpretation of the protocol or method.

**Reviewer Confidence:**

4: Quite sure. I tried to check the important points carefully. It's unlikely, though conceivable, that I missed something that should affect my ratings.

**Typos Grammar Style And Presentation Improvements:**

l. 152 we make use of
l. 166 be it be

---

> ### Author Rebuttal · Authors · 2023-08-29
>
> We thank the reviewer for the comprehensive review and insightful feedback. Regarding the comments about the usage of larger, more recent LLMs like GPT-3.5, we would like to point out that this is an alternative that we naturally considered. However, several reasons let us to the selection of models that appears in the paper, as follows.
>
> (1) Firstly, the decision of using our selected LLMs responds to the necessity to compare on an apples-to-apples manner to previous work, concretely, to the results of Abdou et al. (2021). Since our alignment experiments are an extension of their work, we were interested in confirming if the alignment reported by them still holds on the new settings we propose (a new dataset).
>
> (2) In the second place, we would like to point out that in order to perform our alignment experiments we require to extract embeddings from the LLM, which means that in principle only white-box models can be used. This ruled out API-based systems like GPT-3 and other similar. Moreover, even for competitive white-box models such as LLaMa, given the size of our introduced dataset (~1 million examples), our estimates indicated that embedding efforts would unfortunately lie beyond our budget.
>
> (3) Finally, we note that our selection of LLMs lies within the masked-language modelling domain (MLM). This is a deliberate and critical decision, as it allows for our experiments to be performed in a controlled in-filling setting, limiting what the LLM can output and allowing us to parse their generations automatically. More up-to-date models are all causal LMs (with a few exceptions), which means that our capacity to control is more limited, as these models cannot in-fill text. Moreover, it has been shown that the output of these models is highly dependent on how they are prompted, usually requiring a huge amount of work into prompt construction in order to control the output, which adds further complications.
>
> To illustrate this issue, we implemented a test workflow based on the reviewer's feedback, considering ChatGPT (web-based query) and llama-2 (llama-2-13b-chat). We found that, for the prediction of the comparative task, in several cases (different prompts) even when using k-shot settings, these models often created new types of comparatives (e.g. "SIMILAR IN LIGHTNESS"). As such new comparatives are not present in the ground truth of our dataset, our evaluation framework becomes increasingly complicated since we would need to further post-process the model outputs. Though this lies beyond the scope of our paper, we regard your idea as future work.
>
> We sincerely hope the reviewer can re-consider the score in light of our comments. We are happy to clarify the rationale behind our model selection and clearly explain the limitations of our current approach.
>
> References:
> - Abdou et al. (2021): https://aclanthology.org/2021.conll-1.9.pdf

---

### Official Review · Reviewer_tgng · 2023-08-04

**Soundness:** 4

**Excitement:**

4: Strong: This paper deepens the understanding of some phenomenon or lowers the barriers to an existing research direction.

**Paper Topic And Main Contributions:**

This short paper presents a new (derived) data resource on "naturally occuring" colour names and uses it in two experiments meant to probe the extent to which perceptual structure is mirrored in conceptual structure induced by (text-only) models. In the first experiment, following earlier work, a direct mapping is induced on the data points between the colour space of the named colours and the representation space for the colour names. Unlike in prior work, the resulting alignment is low, which the authors speculate may be due to the more complex names used in this resource. In the second experiment, the linguistic representation space is probed by predicting relations between points ("darker as", "murkier", etc.). Here, it is shown that the predictions for a relation word made by the text-only model largely conform with what the perceptual space would predict.

- NLP engineering experiment
- new data resources
- replication (experiment I)


**Questions For The Authors:**

A: What do you mean by "expected usage" in line 181?


**Reasons To Accept:**

- useful (derived) data resource; cleaned-up subset of ColorNames
- interesting finding, in the failure to replicate direct alignment, but finding good prediction performance on comparatives


**Reasons To Reject:**

- none


**Reproducibility:**

4: Could mostly reproduce the results, but there may be some variation because of sample variance or minor variations in their interpretation of the protocol or method.

**Reviewer Confidence:**

4: Quite sure. I tried to check the important points carefully. It's unlikely, though conceivable, that I missed something that should affect my ratings.

**Typos Grammar Style And Presentation Improvements:**

- l 001: "the need for grounding in language understanding is an active research topic" sounds clunky... "whether language understanding requires grounding is an active topic"?

- l 009: I find it a bit misleading to say that you "collect" this resource -- you derive a useful dataset from an existing dataset that someone else collected.

- l 011: "examples , and"

- l 039: "colors in general is tractable" .. sounds odd grammatically, but also not clear what you mean. Tasks can be tractable or not, but colours aren't a task

- l 166: "be it be using" --> by using?

- l 255: closing ] is missing

- l 268: would be helpful to denote the prompt also typographically, or at least put it into quotes: "is appended as "description is [MASK] than description", i.e. the ..."

- l 281: I guess you forgot to poner el numero aca :-)

---

> ### Author Rebuttal · Authors · 2023-08-29
>
> We thank reviewer `tgng` for the comprehensive and constructive review. We appreciate the confirmation that they found no reasons for rejection in the context of a short paper contribution, and that the topic is relevant.
> Regarding question A on the word `usage`, we would like to clarify that this was an unfortunate typo. That paragraph aimed at describing Figure 2, where we added three example words, but `Figure 3` was referenced incorrectly.
> We appreciate the detailed list of typos and mistakes prepared; we will make sure they are fixed for the camera-ready version upon acceptance.

---

### Meta-Review · Area_Chair_5U6h · 2023-09-18

**Recommendation:** 4

**Metareview:**

This paper analyzes representations of color terms and phrases in text-only models, including the alignment between embedding representations and actual color space, as well as relative representations of two color descriptions. The paper finds that subjectivity and abstractness in color descriptions poses a significant challenge for text-only models. The study is well-motivated and very interesting, but there are two methodological concerns brought by a reviewer.

First, that there is a possible confound that might be artificially lowering alignment quality -- specifically, that some color names don't imply they are being used as color names at all. The authors perform additional experiments to show that when only considering color names that actually include known color terms (e.g., "yellow"), alignments are slightly higher quality than those without known color terms. I encourage the authors to include the analysis in a final version of the paper.

Second, that while the data here was generated by people annotating color hex codes, there is no verification or calculation of agreement that collected descriptions accurately represent their paired hex codes. While I agree this analysis would be out of scope for a short paper, the lack of such analysis should be acknowledged in the paper limitations.

---

### Decision · Program_Chairs · 2023-10-07

**Decision:**

Accept-Findings

**Comment:**

This paper analyzes representations of color terms and phrases in text-only models, including the alignment between embedding representations and actual color space, as well as relative representations of two color descriptions. The paper finds that subjectivity and abstractness in color descriptions poses a significant challenge for text-only models. The study is well-motivated and very interesting, but there are two methodological concerns brought by a reviewer.

First, that there is a possible confound that might be artificially lowering alignment quality -- specifically, that some color names don't imply they are being used as color names at all. The authors perform additional experiments to show that when only considering color names that actually include known color terms (e.g., "yellow"), alignments are slightly higher quality than those without known color terms. I encourage the authors to include the analysis in a final version of the paper.

Second, that while the data here was generated by people annotating color hex codes, there is no verification or calculation of agreement that collected descriptions accurately represent their paired hex codes. While I agree this analysis would be out of scope for a short paper, the lack of such analysis should be acknowledged in the paper limitations.